# Air pollution, physical activity and ischaemic heart disease: a prospective cohort study of interaction effects

Wasif Raza  , Benno Krachler, Bertil Forsberg, Johan Nilsson Sommar

Public Health and Clinical Medicine, Sustainable Health, Umeå University, Umea, Sweden

**Correspondence to**
Wasif Raza; wasif.raza@umu.se

## ABSTRACT

**Objective** To assess a possible interaction effect between physical activity and air pollution on first incidence of ischaemic heart disease (IHD).

**Design** Prospective cohort study.

**Setting** Umeå, Northern Sweden.

**Participants** We studied 34 748 adult participants of Västerbotten Intervention Programme cohort from 1990 to January 2014. Annual particulate matter concentrations ($PM_{2.5}$ and $PM_{10}$) at the participants' residential addresses were modelled and a questionnaire on frequency of exercise and active commuting was completed at baseline. Cox proportional hazards modelling was used to estimate (1) association with physical activity at different levels of air pollution and (2) the association with particulate matter at different levels of physical activity.

**Outcome** First incidence of IHD.

**Results** Over a mean follow-up of 12.4 years, there were 1148 IHD cases. Overall, we observed an increased risk of IHD among individuals with higher concentrations of particles at their home address. Exercise at least twice a week was associated with a lower risk of IHD among participants with high residential $PM_{2.5}$ (hazard ratio (HR) 0.60; 95% CI: 0.44 to 0.82) and $PM_{10}$ (HR 0.55; 95% CI: 0.4 to 0.76). The same beneficial effect was not observed with low residential $PM_{2.5}$ (HR 0.94; 95% CI: 0.72 to 1.22) and $PM_{10}$ (HR 0.99; 95% CI: 0.76 to 1.29). An increased risk associated with higher long-term exposure to particles was only observed among participants that exercised in training clothes at most one a week and among those not performing any active commuting. However, only the interaction effect on HRs for exercise was statistically significant.

**Conclusion** Exercise was associated with a lower risk of first incidence of IHD among individuals with higher residential particle concentrations. An air pollution-associated risk was only observed among those who exercised less. The findings support the promotion of physical activity and a mitigation of air pollution.

## INTRODUCTION

Cardiovascular disease (CVD) is the most important cause of morbidity and premature mortality worldwide, accounting for 422.3 million cases and 17.92 million deaths in 2015.[1] There is solid evidence that inflammation is a key upstream pathogenic mechanism.[2] Ambient air pollution, generally

---

### Strengths and limitations of this study

► This study simultaneously evaluated the impact of physical activity and air pollution and their interaction on first incident ischaemic heart disease cases in a population with relatively low level of air pollution concentrations.

► For air pollution exposure, this study used individual time-varying exposures of annual mean concentrations during follow-up based on population address registries.

► Another strength is the prospective design and the availability of baseline data on several important confounders.

► No exposure–response assessment could be performed since the statistical power only allowed for the formation of two exposure categories.

► Differences in air pollution exposure during active commuting might cause biased estimates due to exposure misclassification.

---

comprising ozone, sulphur dioxide, nitrogen dioxide ($NO_2$) and particulate matter (PM), is a leading contributor to the global burden of disease and an important risk factor for morbidity and mortality. PM is often measured as $PM_{2.5}$, to represent particles with a diameter of 2.5 μm or less, and $PM_{10}$, to represent particles with a diameter of 10 μm or less. Ambient $PM_{2.5}$ exposure alone has been estimated to account for 4.2 million deaths in 2015, of which 1.5 million deaths were caused by ischaemic heart diseases (IHDs).[3] The underlying mechanisms mediating the pathogenic impact of air pollution involve systemic oxidative stress and inflammation.[4 5] Physical activity is a salutogenic factor in numerous non-communicable chronic diseases, with physical inactivity being responsible for 6% of the burden of disease from coronary heart disease.[6 7] The beneficial effects of physical activity include protection against low-grade inflammation by releasing anti-inflammatory substances, such as interleukin 6, from contracting muscles.[8 9]

Rapid urbanisation and increased use of motorised transport contribute to modern-day problems such as traffic congestion, traffic-related air pollution and lack of physical activity. Promotion of active transportation by changing mode of transport from car to cycling and walking are among the different strategies used to tackle these challenges.[10–12] As inflammation is a causative mechanism for CVD, it is conceivable that the anti-inflammatory effects of physical activity may mitigate the harmful effects associated with exposure to air pollution. However, one major concern with physical activity in a polluted environment is the increased inhalation of particles due to an increase in respiratory volume that may counteract the beneficial effects of physical activity.[13 14]

The long-term effects of air pollution among individuals with different levels of leisure time physical activity have been estimated within the Danish Diet, Cancer, and Health cohort. The incidence of diabetes was assessed in relation to leisure time physical activity and $NO_2$ concentration at the home address. Residential $NO_2$ was found to be associated with increased incidence of diabetes, but only among physically active individuals.[15] As far as we know, only one study has examined the modifying effect of air pollution on the association between physical activity and CVD. In their recent study, Kubesch *et al* conclude that physical activity reduced the risk of first incidence of myocardial infarction (MI) and recurrent MI among individuals with high $NO_2$ concentration at the residential addresses.[16] As only one pollutant ($NO_2$) and only one cardiovascular outcome (MI) have been studied, the knowledge of a possible interaction between air pollution and physical activity on CVD is inconclusive.

A Taiwanese study found an independent inverse association of habitual physical activity with inflammation across different levels of $PM_{2.5}$ exposure, although long-term exposure was associated with increased inflammation at all levels of physical activity.[17]

We therefore aimed to examine interaction effects between physical activity and long-term exposure to $PM_{2.5}$ and $PM_{10}$ at residential addresses on the incidence of IHD. We wanted to assess: (1) whether air pollution modifies the beneficial effects of physical activity on IHD and (2) whether physical activity modifies the harmful effects of air pollution on IHD.

## METHODS

To determine the interaction effect between air pollution and physical activity on IHD incidence, we combined cohort data which comprised risk factors for IHD, national registry data on IHD incidence from the Swedish National Board of Health and Welfare, and yearly annual mean air pollution particle concentrations at the individuals' residential addresses using dispersion models from the Swedish Clean Air and Climate Research Program (SCAC).

Västerbotten Intervention Programme (VIP) is a population-based screening and prospective cohort study,

developed to reduce the risk of future CVD and diabetes by promoting a healthy lifestyle among individuals living in the Västerbotten region. The SCAC developed methods to estimate exposure to source-specific PM such as $PM_{2.5}$ and $PM_{10}$ at residential addresses in Gothenburg, Stockholm and Umeå.

## Study population

VIP is an ongoing population-based health investigation survey of all individuals at ages 40, 50 or 60 years, depending on risk factors, living in the Västerbotten region, who are invited to participate in systematic risk factor screening and individual counselling about healthy lifestyle habits. A detailed description of VIP has been presented elsewhere.[18] Between 1990 and 2014, a total of 42 488 of the VIP participants who lived in Umeå Municipality during the study period were included in the analysis. After exclusion of 7740 participants with missing information on exercise, the study sample thus consisted of 34 748 individuals, 53% men and 47% women, 40–60 years of age at baseline examination, with no history of IHD at the time of enrolment. After exclusion of individuals with missing information on included confounders, the final numbers of included individuals were 31 424 and 29 218 for the analyses of exercise in training clothes and active commuting, respectively. All participants in the VIP gave their informed written consent.

## Leisure time exercise and active commuting

The VIP questionnaire includes a lot of self-reported information on physical activity including frequency of leisure time physical activity and active commuting (cycling or walking to and from work). The association with IHD was assessed in relation to frequency of exercise in training clothes and amounts of active commuting. Exercise during the previous 3 months was categorised as never, rarely, once per week, two to three times per week or more than three times per week. Based on this information, participants were categorised as 'Twice per week or more' if they exercised with a frequency of two to three times per week and more, 'At most once a week' if they exercised rarely or once per week, and 'Never' if no activity was performed. For active commuting, participants were asked about their mode of transport to work each season. Participants were classified in three categories: 'Non-active commuting' if commuting every season by car or bus, 'At most two seasons out of four' if cycling or walking at most half a year and 'More than two seasons out of four', if cycling and walking more than half a year.

## Covariates

The VIP questionnaire also gathered information on participants' educational status, occupation, smoking, alcohol intake and economic status. Education was defined according to the International Standard Classification of Education, UNESCO 1997. Participants were asked about the highest level of education they had achieved with eight predefined categories ranging from

'preschool' to 'university education.' Alcohol intake was assessed by the reported frequency of consumption with answering options that ranged from 'never' to 'two to four times/week.' Information on smoking was gathered by using the question, 'How often do you smoke?' Information on occupational status was obtained with a question, 'What kind of job do you have nowadays?' with the answering options of eight predefined categories. Finally, information on occupation status was asked with the question, 'What is your current occupation?' with eight categories ranging from permanently employed to retired.

### Air pollution concentrations

Annual mean total concentrations of $PM_{2.5}$ and $PM_{10}$ for the years 1990–2011 (and thereafter linearly extrapolated up until year 2014) were obtained from the SCAC research programme, described in detail elsewhere.[19] Briefly, concentrations of $PM_{2.5}$ and $PM_{10}$ were estimated within SCAC by applying dispersion models on local or regional emission inventories. These emission inventories contain detailed information on emissions from different source categories, such as road traffic exhaust, road traffic non-exhaust, domestic heating, shipping and industrial activities. For Umeå, inventories were validated through monitoring for consistency. Emissions from small-scale residential heating were assessed using registry data from chimney sweepers that included the type of wood, stove or boiler. The addresses of these residences were geocoded using the geographical centre coordinate of the estate. Road traffic emission factors for PM exhaust for different vehicle types, speeds and driving conditions were calculated based on the Handbook Emission Factors for Road Transport V.3.1.[20] Estimates for non-exhaust contribution from brake and tire wear were based on Omstedt *et al*.[21] The annual average emission from shipping was used in the modelling on a $1 \times 1 \, km^2$ grid resolution. The emissions from other sources such as industrial processes, off-road machinery and agriculture were collected from the Swedish Meteorological and Hydrological Institute (SMHI). To obtain annual average emissions of $PM_{2.5}$ and $PM_{10}$, Gaussian models included in the Airviro air quality management system SMHI, 2010 were used for simulation based on hourly meteorological data for 1990, 2000 and 2011.[22] The comparison between measured and modelled $PM_{2.5}$ and $PM_{10}$ agreed well at most monitoring stations ($r^2 = 0.87$ and $r^2 = 0.65$, respectively).

### Outcome

We linked the records in VIP and through the unique Swedish personal identification number with data on first IHD event cases from the National Patient Register and the Cause of Death Register, both at the National Board of Health and Welfare, using primary discharge diagnoses for IHD according to the International Classification of Diseases, 10th revision: code I20–I25.

### Patient and public involvement

No patients were involved in this study.

### Statistical analysis

We performed survival analyses using Cox regression proportional hazards model to estimate HRs and 95% CIs, to estimate (1) the association between first incident IHD and air pollution exposure at different levels of physical activity, and (2) the association between first incident IHD and physical activity at different levels of air pollution exposure. Age was used as the underlying timescale since it is a stronger confounder than calendar time. Follow-up started at date of recruitment to the cohort and ended with the earliest of the date of first IHD case, emigration, death or 31 December 2013. Interaction between physical activity and air pollution and their impact on IHD was studied by introducing an interaction term into the model. Residential annual mean particle concentrations were used to calculate moving averages over the recent 5 years which were thereafter categorised as below or above the median concentration for $PM_{2.5}$ and $PM_{10}$, respectively. Sensitivity analyses were also performed with PM concentrations categorised by tertiles. Interaction with physical activity was assessed based on (1) the frequency of exercise in training clothes and (2) the number of seasons the individual walked or cycled to work. Active commuting by walking or cycling was categorised into three groups: non-active commuters, active commuters at most half of the year (up to two out of four seasons), active commuters more than half a year (more than two out of four seasons). Estimates were adjusted for a prespecified set of covariates: calendar year as a penalised cubic spline with 3 df, gender (male vs female), highest education level (compulsory, high school, university), alcohol intake (never, once/month or sometimes, 2–4 times/month, 2–3 times/week, ≥4 times/week), smoking (previous non-regular smoker, non-regular smoker, cigarette smoker, cigar or pipe smoker), occupation (gainfully employed, unemployed, not gainfully employed, retired), and registry data on area level mean income year 1994. In the basic model, we adjusted only for gender and exposure year. All analyses were performed using R V.3.4.2, and the statistical inference was conducted with a 5% significance level. T-tests, global analysis of variance tests and $X^2$ tests were used to test for differences in means and proportions of covariates between categories of exercise in training clothes (table 1). The Schoenfeld residuals test was used to assess the assumption of proportional hazards. A sensitivity analysis was conducted by excluding participants with follow-up time below the 25th percentile.

## RESULTS

The mean age at recruitment to this cohort was 45.8 years. Among the 34 748 participants, 1148 cases of IHD were identified during a mean follow-up time of 12.4 years. Of those cases, 500 never exercised, 529 exercised at most once a week and 119 exercised at least twice a week. Table 1 summarises characteristics of participants according to different levels of leisure time

**Table 1** Characteristics of participants at different levels of exercise in training clothes at baseline

| Characteristics | Never | At most once a week | Twice per week or more | p-value |
|---|---|---|---|---|
| Exercise in training clothes, n (%) | 13 043 (37.5) | 14 994 (43.2) | 6711 (19.3) | |
| $PM_{10}$, µg/m$^3$ (mean (SD)) | 10.93 (2.00) | 11.06 (1.96) | 10.39 (1.94) | <0.001 |
| $PM_{2.5}$, µg/m$^3$ (mean (SD)) | 6.41 (1.12) | 6.49 (1.12) | 6.11 (1.11) | <0.001 |
| Frequency of active commuting (%) | | | | <0.001 |
| Non-active commuting | 6369 (48.8) | 6409 (42.7) | 2666 (39.7) | |
| At most two seasons of four | 1806 (13.8) | 2547 (17.0) | 1037 (15.5) | |
| More than two seasons of four | 3506 (26.9) | 4976 (33.2) | 2555 (38.1) | |
| Missing | 1362 (10.4) | 1062 (7.1) | 453 (6.8) | |
| Alcohol intake (%) | | | | <0.001 |
| Never | 135 (1.0) | 120 (0.8) | 79 (1.2) | |
| Once/month or sometimes | 5337 (40.9) | 5950 (39.7) | 2790 (41.6) | |
| 2–4 times/month | 2000 (15.3) | 2436 (16.2) | 1061 (15.8) | |
| 2–3 times/week | 97 (0.7) | 93 (0.6) | 54 (0.8) | |
| ≥4 times/week | 406 (3.1) | 534 (3.6) | 462 (6.9) | |
| Missing | 5068 (38.9) | 5861 (39.1) | 2265 (33.8) | |
| Smoking (%) | | | | <0.001 |
| Never smoker | 5139 (39.4) | 7240 (48.3) | 3467 (51.7) | |
| Previous non-regular smoker | 1072 (8.2) | 1458 (9.7) | 711 (10.6) | |
| Non-regular smoker | 581 (4.5) | 807 (5.4) | 348 (5.2) | |
| Previous regular smoker | 2729 (20.9) | 2865 (19.1) | 1201 (17.9) | |
| Cigarette smoker | 2789 (21.4) | 1805 (12.0) | 426 (6.3) | |
| Cigar or pipe smoker | 186 (1.4) | 130 (0.9) | 45 (0.7) | |
| Missing | 547 (4.2) | 689 (4.6) | 513 (7.6) | |
| Highest education level (%) | | | | <0.001 |
| Compulsory | 5534 (42.4) | 4319 (28.8) | 1369 (20.4) | |
| High | 3573 (27.4) | 4116 (27.5) | 1951 (29.1) | |
| University | 3449 (26.4) | 5933 (39.6) | 2911 (43.4) | |
| Missing | 487 (3.7) | 626 (4.2) | 480 (7.2) | |
| Gender %, (men) | 7072 (54.2) | 8036 (53.6) | 3274 (48.8) | <0.001 |
| Age, years (mean (SD)) | 47.3 (9.1) | 45.3 (9.0) | 44.3 (8.6) | <0.001 |
| Occupation (%) | | | | <0.001 |
| Gainfully employed | 10 536 (80.8) | 12 690 (84.6) | 5452 (81.2) | |
| Unemployed | 478 (3.7) | 382 (2.5) | 174 (2.6) | |
| Not gainfully employed | 318 (2.4) | 352 (2.3) | 160 (2.4) | |
| Retired | 673 (5.2) | 449 (3.0) | 194 (2.9) | |
| Missing | 1038 (8.0) | 1121 (7.5) | 731 (10.9) | |
| Mean income for the neighbourhood (SEK) (mean (SD)) | 128 286 (23 018) | 130 332 (23 875) | 130 222 (24 606) | <0.001 |

SEK, Swedish krona.

physical activity. Participants not reporting any leisure time exercise were older and more likely to be men, non-commuters and to belong to a lower socioeconomic group. Subjects performing moderate to high-level physical activity were more likely to be women, non-smokers and active commuters. The 5-year means of $PM_{10}$ and $PM_{2.5}$ concentrations were different between leisure time physical activity categories, with at most 7% and 6%

difference, respectively, for $PM_{10}$ and $PM_{2.5}$ (with distributions presented in online supplemental figures 1–4).

Compared with individuals who reported no exercise, those participants who exercised at least twice per week had a 24% lower risk of IHD (table 2). The corresponding overall estimate associated with active commuting was a 13% reduced risk of IHD among individuals commuting more than two seasons per year.

**Table 2** Hazard ratios (HRs (95% CI)) for IHD associated with different exercise and commuting habits among persons with different air pollution exposure at home addresses

| Exercise in training clothes | Overall model with no interaction effects | Proportional hazard p-value* | Adjusted† HRs in categories of high and low particle exposure | | | | Adjusted† interaction HR | |
|---|---|---|---|---|---|---|---|---|
| | | | Low PM₁₀‡ | Proportional hazard p-value* | High PM₁₀‡ | Proportional hazard p-value* | Benefits of exercise/commuting comparing high and low particle exposure | Proportional hazard p-value* |
| Never | 1 | | 1 | | 1 | | 1 | |
| ≤Once/week | 1.03 (0.90 to 1.16) | 0.35 | 1.01 (0.84 to 1.21) | 0.67 | 1.04 (0.88 to 1.23) | 0.35 | 1.03 (0.81 to 1.32) | 0.75 |
| ≥Twice/week | 0.76 (0.62 to 0.93) | 0.08 | 0.99 (0.76 to 1.29) | 0.47 | 0.55 (0.40 to 0.76) | 0.24 | 0.56 (0.37 to 0.84) | 0.65 |
| | | | Low PM₂.₅§ | | High PM₂.₅§ | | | |
| Never | | | 1 | | 1 | | 1 | |
| ≤Once/week | | | 1.03 (0.86 to 1.23) | 0.78 | 1.03 (0.87 to 1.22) | 0.30 | 1.00 (0.78 to 1.28) | 0.61 |
| ≥Twice/week | | | 0.94 (0.72 to 1.22) | 0.69 | 0.60 (0.44 to 0.82) | 0.09 | 0.64 (0.43 to 0.96) | 0.30 |
| Active commuting per season | | | Low PM₁₀‡ | | High PM₁₀‡ | | | |
| Non-active commuting | 1 | | 1 | | 1 | | 1 | |
| ≤Two seasons of four | 1.01 (0.86 to 1.19) | 0.50 | 1.17 (0.93 to 1.47) | 0.35 | 0.88 (0.70 to 1.11) | 0.70 | 0.76 (0.55 to 1.04) | 0.34 |
| >Two seasons of four | 0.87 (0.76 to 0.998) | 0.85 | 0.93 (0.76 to 1.14) | 0.48 | 0.82 (0.68 to 0.98) | 0.89 | 0.88 (0.67 to 1.15) | 0.53 |
| | | | Low PM₂.₅§ | | High PM₂.₅§ | | | |
| Non-active commuting | | | 1 | | 1 | | 1 | |
| ≤Two seasons of four | | | 1.18 (0.94 to 1.49) | 0.43 | 0.87 (0.69 to 1.10) | 0.76 | 0.73 (0.53 to 1.01) | 0.43 |
| >Two seasons of four | | | 0.97 (0.80 to 1.19) | 0.37 | 0.79 (0.65 to 0.95) | 0.89 | 0.81 (0.62 to 1.06) | 0.44 |

*P-value of the Schoenfeld residual test of proportional hazards.
†Adjusted for sex, calendar year, education, smoking, alcohol intake, occupation, neighbourhood mean income, leisure time physical activity and active commuting.
‡Low PM₁₀: ≤9.6 µg/m³, high PM₁₀: >9.6 µg/m³.
§Low PM₂.₅: ≤5.7 µg/m³, high PM₂.₅: >5.7 µg/m³.
IHD, ischemic heart disease.

Allowing for an interaction between the frequency of exercise in training clothes and particle concentrations ($PM_{10}$ and $PM_{2.5}$) at the home address, the average 24% risk reduction from exercising at least twice per week was found to be driven by statistically significant interaction between exercise and particle exposure with 45% and 40% risk reduction among individuals with high $PM_{10}$ and $PM_{2.5}$ concentrations, respectively (table 2). The interaction coefficients estimating the additional benefit of exercise among individuals with a high $PM_{10}$ concentration at their home addresses were a 3% increased risk among those who exercised at most once a week, whereas a decreased risk of 44% was estimated among those who exercised at least twice a week. The corresponding estimates among those with high $PM_{2.5}$ concentrations were risk reductions of 0% and 36%, respectively.

For active commuters with low particle exposure at their home address, the risks of incident IHD were 17% and 18% higher among those commuting one or two seasons per year and 7% and 3% lower among those commuting at least two seasons per year, for $PM_{10}$ and $PM_{2.5}$, respectively. The benefit of active commuting was larger among individuals with a high particle concentration at their home address: risk reductions for active commuting during one or two seasons were 12% and 13% and for more than two seasons, 18% and 21%, respectively, compared with non-active commuters. No statistically significant interaction was found between active commuting and particle concentrations at the home address.

Individuals exposed to high concentrations of $PM_{10}$ and $PM_{2.5}$ at their home address had a 14% and 1% increased risk of incident IHD, respectively, compared with individuals with low concentrations (table 3). These increased risks were, however, not statistically significant.

When including an interaction between particle concentrations and exercise, risk estimates showed a positive association between air pollution and IHD among individuals performing no exercise and those exercising no more than once a week, whereas a negative association was found among those exercising more than twice a week (table 3).

Compared with individuals with low residential particle concentration, a high concentration of $PM_{10}$ was associated with 21% and 25% increased risk of IHD for those who never exercised in training clothes and those who exercised at most once a week, respectively, and showed a 32% decreased risk for those exercising at least twice a week; the association for those who never exercised was statistically non-significant. The corresponding estimates associated with $PM_{2.5}$ are increased risk of IHD of 16% for those never exercising and those who exercised at most once a week, and a decreased risk of 26% among those exercising at least twice a week (table 3); none of these associations was statistically significant.

IHD risk associated with high residential $PM_{10}$ and $PM_{2.5}$ compared with low residential particle concentration was found to be 26% and 24% higher, respectively, among those never actively commuting; both these

results were statistically significant. Among those actively commuting one or two seasons per year, IHD risks were 5% and 9% lower, respectively, while among those actively commuting more than two seasons out of four, the risks were 10% and 1% higher, respectively; none of these risks were, however, statistically significantly different from the air pollution-associated risks among the active commuters (table 2). Overall, no statistically significant modifying effect of active commuting on the association between high particle concentration at home addresses and IHD was observed (table 3).

### Sensitivity analyses

Excluding participants with short follow-up time (below the 25th percentile of 3.4 years) did not affect the main conclusions of our study, however estimates tended to be lower for overall effect of air pollution on IHD (online supplemental tables 1 and 2).

Sensitivity analyses were also conducted with PM concentrations in tertiles. An indication of a dose–response was found, with increasing benefits of exercise in training clothes with higher levels of PM concentrations at the home address (online supplemental table 3). Exercising at least twice per week (compared with never) reduced the risk of incident IHD by 5, 17 and 49% within the first, second and third tertile of $PM_{10}$ exposure, respectively. Similar risk reductions were found in relation to tertiles of $PM_{2.5}$.

No such interaction dose–response was however found for risks associated with PM exposure. Risk estimates associated with PM were somewhat higher among individuals who exercised once a week compared with those who never exercised, whereas no increased risk associated with either $PM_{10}$ or $PM_{2.5}$ was found among individuals who exercised at least twice per week (online supplemental table 4). A dose–response with increasing risks for IHD were found with both $PM_{10}$ and $PM_{2.5}$ among those who never exercised.

These interactions between PM concentrations and exercise at least twice per week (compared with never) were statistically significant for high $PM_{10}$ and borderline statistically significant for high $PM_{2.5}$ (online supplemental table 5).

### DISCUSSION

Overall, we found increased risk of first incident IHD associated with air pollution at the home address but a protective effect of physical activity. A statistically significant beneficial effect of exercise was found among individuals with high $PM_{10}/PM_{2.5}$, but not among individuals with low levels. Also, for active commuting, the benefits were greater among individuals with high residential particle concentrations, but these differences were not statistically significant. Air pollution concentration-associated risks were found among individuals who exercise at most once a week but not among individuals exercising at least twice a week. Statistically significantly increased risks were also

Table 3 Hazard ratios (HRs (95% CI)) for IHD associated with high air pollution levels (vs low) at home address among persons with different exercise/commuting habits

| | Overall model with no interaction effects | Proportional hazard p value* | Adjusted† HRs in categories of high and low particle exposure | | | | | |
|---|---|---|---|---|---|---|---|---|
| | | | **Exercise in training clothes** | | | | | |
| | | | **Never** | **Proportional hazard p-value*** | **≤Once/week** | **Proportional hazard p-value*** | **≥Twice/week** | **Proportional hazard p-value*** |
| Low $PM_{10}$‡ | 1 | | 1 | | 1 | | 1 | |
| High $PM_{10}$‡ | 1.14 (0.9 to 1.45) | 0.55 | 1.21 (0.97 to 1.49) | 0.71 | 1.25 (1.01 to 1.54) | 0.45 | 0.68 (0.46 to 0.998) | 0.50 |
| Low $PM_{2.5}$§ | 1 | | 1 | | 1 | | 1 | |
| High $PM_{2.5}$§ | 1.01 (0.8 to 1.28) | 0.3 | 1.16 (0.94 to 1.44) | 0.38 | 1.16 (0.95 to 1.43) | 0.13 | 0.74 (0.51 to 1.09) | 0.11 |
| | | | **Active commuting** | | | | | |
| | | | **Non-active commuting** | | **≤Two seasons of four** | | **>Two seasons of four** | |
| Low $PM_{10}$‡ | | | 1 | | 1 | | 1 | |
| High $PM_{10}$‡ | | | 1.26 (1.03 to 1.54) | 0.39 | 0.95 (0.7 to 1.29) | 0.66 | 1.10 (0.86 to 1.41) | 0.17 |
| Low $PM_{2.5}$§ | | | 1 | | 1 | | 1 | |
| High $PM_{2.5}$§ | | | 1.24 (1.02 to 1.51) | 0.11 | 0.91 (0.67 to 1.24) | 0.84 | 1.01 (0.79 to 1.29) | 0.04 |

*p-value of the Schoenfeld residual test of proportional hazards.
†Adjusted for sex, calendar year, education, smoking, alcohol intake, occupation, neighbourhood mean income, leisure time physical activity and active commuting.
‡Low $PM_{10}$: ≤9.6 µg/m$^3$; high $PM_{10}$: >9.6 µg/m$^3$.
§Low $PM_{2.5}$: ≤5.7 µg/m$^3$; high $PM_{2.5}$: >5.7 µg/m$^3$.
IHD, ischemic heart disease.

found among non-active commuters. These risks were, however, not statistically significantly different from air pollution-associated risks among the active commuters.

Our findings are in accordance with a longitudinal cohort study on MI within the Danish Diet, Cancer, and Health cohort which found an increased benefit of participation in sports among individuals with high $NO_2$ concentration at the home addresses.[16] The reported risk reduction was 9, 15, and 24%, respectively, among individuals with low (<14.3 µg/m$^3$), medium (14.3–21 µg/m$^3$) and high (>21 µg/m$^3$) residential $NO_2$ concentration. For walking and cycling, they also estimated greater risk reductions for first incident MI among individuals with higher air pollution concentrations. In the same Danish cohort, the long-term benefits of physical activity on CVD mortality were also found to be greater among individuals with high residential $NO_2$.[23] The risk reduction associated with participation in cycling and gardening among individuals exposed to high residential $NO_2$ (≥19 µg/m$^3$) was greater than those exposed to moderate/low $NO_2$ concentration (<19 µg/m$^3$). Among participants exposed to high $NO_2$, the risk reductions for cycling and gardening were 30% and 23%, respectively; whereas among participants exposed to low $NO_2$, risk reductions were 17% and 15%, respectively. However, the interaction effects in these two studies were not statistically significant.

Opposite findings were observed for the modifying effect of physical activity on the association between air pollution and the incidence of diabetes.[15] Among the participants in the Danish cohort, the risk of developing diabetes increased by 10% per IQR of 4.9 mg/m$^3$ residential $NO_2$ among physically active individuals, but there was no difference among less physically active individuals. The authors considered that this may be due to an additive rather multiplicative interaction, thus resulting in an increased risk estimate only among physically active individuals with a low risk of developing diabetes.

As inflammation is a causative mechanism for CVD, we hypothesised that the anti-inflammatory effects of physical activity may reduce air pollution-associated risks since inflammation is one among several different pathways for the harmful health effects of air pollution. The findings of greater benefits of physical activity among individuals with higher air pollution exposure for incident IHD risk in our study and incident MI and CVD mortality in the Danish cohort support such a hypothesis. However, a study on physical activity and white cell counts conducted in a large cohort of Taiwanese adults suggested no effect modification by residential air pollution measured as $PM_{2.5}$.[17] Both physical activity and residential air pollution were, however, found to be associated with an inflammatory response assessed by white cell counts. However, the association between physical activity and white cell count is variable because exercise also causes a transient increase in white cell count which usually normalises within 24 hours.[24] The results of our study cannot be directly compared with the above-mentioned studies due to the difference in pollutants[15 16 23] and health outcomes.[15 16 23]

This study contributes to air pollution effect estimates on IHD incidence in a population with a relatively low level of air pollution concentrations. Compared with the previous cohort studies on interaction effects between air pollution and physical activity, the annual mean $PM_{2.5}$ concentration was three to four times lower. The annual mean in the Taiwanese cohort was 27 µg/m$^3$ and a recent study within the same population as the Danish Diet, Cancer, and Health cohort studies reported 18 µg/m$^3$. Even at these lower levels of air pollution, an increased risk associated with air pollution exposure was found, however not among those who exercised at least twice a week. Even though active commuting may result in higher air pollution exposure compared with, for instance, driving a car to work, the risk of an IHD event was still reduced since the benefit of the physical activity was greater than the IHD risk imposed by the air pollution exposure.

A major strength of our study is the air pollution particle concentration exposure data since particles are considered to be the causal component of air pollution.[25] The study used individual time-varying exposures of annual mean concentrations during follow-up based on population address registries. The dispersion model used for modelling of particle concentrations has previously been validated.[19] Within the DHC studies, $NO_2$ was used as a proxy for traffic-generated air pollution and was assessed only at residential addresses at the year of recruitment or as an annual mean during follow-up. A limitation of our exposure data is that the statistical power only allowed for two exposure categories and therefore no exposure–response assessment was performed.

Other strengths of our study are the prospective design, the long follow-up period, the large cohort size and the availability of baseline data on several important confounders. A limitation is the lack of information on the intensity and duration of physical activity and therefore only frequency of exercise could be considered. The study also lacked information on changes in physical activity and other lifestyle factors during follow-up as the information was only retrieved at baseline.

There is a risk of reverse causation if individuals at their baseline examination had a low physical activity level due to poorer health. Individuals could, for instance, have diseases that affect their risk to later in life have an IHD event (such as diabetes) prior to baseline examination. If this prior disease also affected the frequency the individual exercised in training clothes, or mode of commuting, then a reverse causation between physical activity and IHD risk may occur. Individuals with a prior IHD event at baseline were however excluded, and the sensitivity analyses that excluded individuals with follow-up time below the 25th percentile showed that this did not change the results. Furthermore, we lacked information on whether exercise is taking place outdoors or indoors. For active commuters, we also lack air pollution-exposure calculations during the commute. This would cause exposure misclassification among active commuters with a higher in-traffic air pollution exposure dose compared with

non-active commuters, causing a possible bias to the null. This would also occur if individuals chose not to exercise outside during times with high air pollution exposure.

## CONCLUSION

The study estimated that exercise reduced the risk of first incident IHD, but only among individuals with higher residential particle concentrations (above median). Similarly, the harmful air pollution effect on IHD was only found among those who exercised less. Our results reinforce the public health message that physical activity is beneficial for cardiovascular health and thus support the adoption of strategies to improve health through promotion of physical activity and mitigation of air pollution. Further studies are needed to build on the evidence of physical activity and air pollution interactions on the incidence of CVD. Air pollution exposure during commuting should also be considered in these studies.

**Contributors** All the authors contributed to the study conception and design. WR and JS analysed the data. WR wrote the first draft of the manuscript. BF, BK and JS critically revised the manuscript for important intellectual content. All authors read and approved the final version of the manuscript.

**Funding** The study was funded by FORTE (Forskingsrådet för hälsa, arbetsliv och välfärd) (grant number: 2012-1296), a Swedish state agency that finances scientific research in health, welfare and working life, and by grants from Folksam research foundation and the Swedish state under the agreement between the Swedish government and the county councils (ALF) awarded to BF (grant number: VLL-645781).

**Competing interests** None declared.

**Patient consent for publication** Not required.

**Ethics approval** The study was approved by the Regional Ethics Review Board at Umea University (DNR: 2014-136-32M and 2015/16-31Ö).

**Provenance and peer review** Not commissioned; externally peer reviewed.

**Data availability statement** No data are available. No additional data are available.

**ORCID iDs**
Wasif Raza http://orcid.org/0000-0002-9420-2433
Johan Nilsson Sommar http://orcid.org/0000-0002-8854-498X

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
