## [Reviewer comments · BMJ Open]

ARTICLE DETAILS

TITLE (PROVISIONAL)	Air pollution, physical activity and ischemic heart disease - A prospective cohort study of interaction effects
AUTHORS	Raza, Wasif; Krachler, Benno; Forsberg, Bertil; Sommar, Johan

VERSION 1 – REVIEW

REVIEWER	Jonathan Yap NHCS, Singapore
REVIEW RETURNED	30-Jun-2020

GENERAL COMMENTS	Dear Editor, The authors have submitted a well-written article on the effects of air pollution and physical activity on ischemic heart disease. They found a beneficial effect of exercise on reducing IHD in more polluted areas, with air pollution impacting those who exercised less. There are a few comments that need to be addressed. Comments 1. This is an important comment that needs to be addressed. The analysis of PM 2.5 and PM 10 could be further looked into.a) The annual concentrations from 2011-2014 were extrapolated. The authors need to explain why this was so. They mention that “The comparison between measured and modelled PM2.5 and PM10 agreed well at most monitoring stations ($r^2=0.87$ and $r^2=0.65$, respectively)”. How many stations did this not agree with? A r^2 of 0.65 does not appear to be very good agreement.b) How was the exposure to PM 2.5 and PM 10 calculated per individual? Did the authors sum up the individual years of PM 2.5/10 and average it? Was the number of years taken from recruitment to event/end of study or a general 5 years for all?c) The authors should provide a distribution of PM 2.5/10 for the study and maybe by regions. The definition of low PM10 was $<9.6\mu\text{g}/\text{m}^3$ and high PM 10 $>9.6\mu\text{g}/\text{m}^3$. This cut-off seems very low and indicates hardly any pollution. This applies also to PM 2.5. Perhaps the authors could analyse PM levels by possibly quartiles and maybe as a continuous variable.2. The authors mention “Interaction between physical activity and air pollution and their impact on IHD was studied by introducing an interaction term into the model”. Was this interaction term significant?3. As the study was performed in those middle aged and above, one major limitation is that those who exercise less at this stage may intrinsically have already poorer baseline health. The authors
---

	have briefly mentioned this in the limitation but should expound more. 4. Those who actively commute may have greater exposure to air pollution but they were noted to have lesser risk of IHD. Similarly for exercise (although data on indoor or outdoor exercise was not available). The authors should comment on this in the discussion. 5. The authors mention “Overall, we observed an increased risk of IHD among physically active individuals with higher concentrations of particles at their home address.” From Table 3, the PM concentration did not seem to impact IHD risk in those who exercised frequently.
--	--

REVIEWER	Neil Wright University of Oxford, UK.
REVIEW RETURNED	20-Nov-2020

GENERAL COMMENTS	The paper describes an investigation of associations between exercise and pollution exposure and incident IHD. The study finds a protective association of physical activity with IHD among those with "high" pollutant exposures. Methods and results are generally well described and clearly presented. Cox proportional hazards models are suitably used to estimate hazard ratios. Interaction terms are used to investigate possible interactions between exercise and pollution exposures. Age is used as the timescale, so that age is adjusted for, however it is not clear that there is appropriate consideration of possible age cohort effects. Annual pollution levels are used as time-varying exposures. The authors note that power limited categorisation of exposure to two categories, defined by the median level. However, the authors do not justify this choice of threshold. A different categorisation for "high" may be more appropriate (e.g. based on existing guidance for pollutant levels), and also aid interpretation. The study would benefit from clear justification of categorisation, sensitivity investigation of the robustness of results to choice of grouping, and/or use of pollution exposure as a continuous variable (using flexible models for shape of association, as required). Additionally, by using a time-varying covariate the study has not investigated or accounted for any cumulative (or carry over) effects of pollutant exposure (or interaction between pollution exposure and exercise). Exercise in training clothes and commuting are each categorised into three groups based on frequency and number of seasons, respectively. These are reported by participants at baseline only, and authors should report this limitation of exposure information. As the authors note in limitations, there is also no information on intensity or duration of physical activity. If possible, investigation of some measure of total physical activity would also be useful. Models are adjusted for calendar year, gender, education, alcohol, smoking, occupation, and area mean income. The authors note that these may be confounders, but do not report clear methods for identifying and selecting confounders. Summary statistics by air pollution category (or associations between covariate and air pollution) would be a useful addition.
---

	The authors do not describe methods or results for assessment of the proportional hazards assumption. These should be carried out and reported, to justify the use of proportional hazards models. As the authors note, there is a risk of reverse causation. Sensitivity analyses could be included to investigate this further (e.g. by excluding an initial follow-up period from follow up). Some further points: Table 1: Report the statistical test used to produce the P values. Lines 306-309: It would be helpful to readers to clarify in the text to which exercise group each number refers. Report the number of IHD events in each exposure group, as well as the overall number, Lines 374-381: This paragraph repeats the results described in lines 364-368. Lines 382-384: This text repeats the results described in lines 370-373. Table 3: CI limits should be reported consistently, with 2 decimal places. I appreciate the authors may wish to show that an interval does not quite include one, but there is no different interpretation/conclusion between an interval that ends at 0.998 or 1.00.
--	--

VERSION 1 – AUTHOR RESPONSE

Reviewer 1

The authors have submitted a well-written article on the effects of air pollution and physical activity on ischemic heart disease. They found a beneficial effect of exercise on reducing IHD in more polluted areas, with air pollution impacting those who exercised less. There are a few comments that need to be addressed.

We thank the Reviewer for the positive appraisal.

1. This is an important comment that needs to be addressed. The analysis of PM 2.5 and PM 10 could be further looked into.

a) The annual concentrations from 2011-2014 were extrapolated. The authors need to explain why this was so. They mention that “The comparison between measured and modelled PM2.5 and PM10 agreed well at most monitoring stations ($r^2=0.87$ and $r^2=0.65$, respectively)”. How many stations did this not agree with? A r^2 of 0.65 does not appear to be very good agreement.

Annual average particle concentrations were only modelled for the years 1990 to 2011 and therefore were extrapolated linearly from 2011 to 2014. Since the main sources behind spatial exposure contrasts are roads and house heating boilers or stoves they do not change over 3 years, only weather and the common regional background levels vary.

In Umeå there are only two permanent stations, but the modelling has been validated within the Swedish Clean Air and Climate (SCAC) program using also Gothenburg and Stockholm. We agree with the reviewer that an R^2 of 0.65 for PM10 does not appear to be very good agreement. This is likely because the coarse fraction of PM10 in Sweden includes much road dust from the use of

studded winter tyres. The agreement is regarded as good for the fine fraction, usually seen as more relevant for health effects.

b) How was the exposure to PM 2.5 and PM 10 calculated per individual? Did the authors sum up the individual years of PM 2.5/10 and average it? Was the number of years taken from recruitment to event/end of study or a general 5 years for all?

The particle exposure for each individual was calculated as moving averages of annual mean concentrations over the recent five years (Lines 250-251)

c) The authors should provide a distribution of PM 2.5/10 for the study and maybe by regions. The definition of low PM10 was $<9.6\mu\text{g}/\text{m}^3$ and high PM 10 $>9.6\mu\text{g}/\text{m}^3$. This cut-off seems very low and indicates hardly any pollution. This applies also to PM 2.5. Perhaps the authors could analyse PM levels by possibly quartiles and maybe as a continuous variable.

The distributions of PM2.5 and PM10 have been added to the supplementaryfiles B1 and B2. A dichotomisation of the particle concentrations was used to achieve sufficient statistical power. We agree with the reviewer that particle concentration level seems low, however associations with incidence of CVD and mortality are in several studies from North America and Europe reported at these levels. We could not analyse particle levels as quartiles and continuous variables due to limited statistical power.

2. The authors mention "Interaction between physical activity and air pollution and their impact on IHD was studied by introducing an interaction term into the model". Was this interaction term significant? We observed statistically significant interaction effects between exercise at least twice a week and high PM10/ PM2.5 and this has now been clarified in the results section (line 300, with interaction effect estimates and confidence intervals presented in Table 2)

3. As the study was performed in those middle aged and above, one major limitation is that those who exercise less at this stage may intrinsically have already poorer baseline health. The authors have briefly mentioned this in the limitation but should expound more.

A more in-depth discussion of this limitation has now been added to the discussion (Lines 466-470).

4. Those who actively commute may have greater exposure to air pollution but they were noted to have lesser risk of IHD. Similarly for exercise (although data on indoor or outdoor exercise was not available). The authors should comment on this in the discussion.

We thank the reviewer for pointing this out and have now commented on this in discussion (Lines 447-450)

5. The authors mention "Overall, we observed an increased risk of IHD among physically active individuals with higher concentrations of particles at their home address." From Table 3, the PM concentration did not seem to impact IHD risk in those who exercised frequently.

We thank the reviewer for pointing out this mistake, which now has been corrected. The sentence should have been: "Overall, we observed an increased risk of IHD among individuals with higher concentrations of particles at their home address." (Line 44)

Reviewer 2

The paper describes an investigation of associations between exercise and pollution exposure and incident IHD. The study finds a protective association of physical activity with IHD among those with "high" pollutant exposures. Methods and results are generally well described and clearly presented. Cox proportional hazards models are suitably used to estimate hazard ratios. Interaction terms are used to investigate possible interactions between exercise and pollution exposures.

We thank the Reviewer for positive appraisal

Age is used as the timescale, so that age is adjusted for, however it is not clear that there is appropriate consideration of possible age cohort effects.

Age was used as the underlying timescale since it is a stronger confounder than calendar time (Lines 245-246). In the model, the time trend in age-specific incidence of IHD was modelled by fitting a smooth function (penalized cubic spline with 3 degrees of freedom). Any problematic cohort effect is not foreseen since there has been a continuous recruitment into the cohort during the study period. Annual pollution levels are used as time-varying exposures. The authors note that power limited categorisation of exposure to two categories, defined by the median level. However, the authors do not justify this choice of threshold. A different categorisation for "high" may be more appropriate (e.g. based on existing guidance for pollutant levels), and also aid interpretation. The study would benefit from clear justification of categorisation, sensitivity investigation of the robustness of results to choice of grouping, and/or use of pollution exposure as a continuous variable (using flexible models for shape of association, as required). Additionally, by using a time-varying covariate the study has not investigated or accounted for any cumulative (or carry over) effects of pollutant exposure (or interaction between pollution exposure and exercise).

The median particle concentration was used as cut-off to obtain highest statistical power. To base the categorization on guidance or official limit values is not possible since concentrations in our study area are below these values. There is also no lower limit of exposure for which no health effects occur. Using continuous exposures was further complicated by a higher correlation with covariates (for instance exercise habits). Exposures were time-varying but were assessed as five-year moving averages and therefore considers the cumulative IHD risk associated with this exposure.

Exercise in training clothes and commuting are each categorised into three groups based on frequency and number of seasons, respectively. These are reported by participants at baseline only, and authors should report this limitation of exposure information.

A clarification of this limitation has been added to the discussion (Line 464).

As the authors note in limitations, there is also no information on intensity or duration of physical activity. If possible, investigation of some measure of total physical activity would also be useful. We agree with the reviewer that a total physical activity measure would be useful. However, we could not retrieve complete information on occupational physical activity.

Models are adjusted for calendar year, gender, education, alcohol, smoking, occupation, and area mean income. The authors note that these may be confounders, but do not report clear methods for identifying and selecting confounders. Summary statistics by air pollution category (or associations between covariate and air pollution) would be a useful addition.

We selected the confounders which were consistent with previous studies and known to affect the association between air pollution, physical activity and IHD.

The authors do not describe methods or results for assessment of the proportional hazards assumption. These should be carried out and reported, to justify the use of proportional hazards models.

P-values for the Schoenfeld test of proportional hazards have been added to the result section (Lines 264-265, and Tables 2 and 3).

As the authors note, there is a risk of reverse causation. Sensitivity analyses could be included to investigate this further (e.g. by excluding an initial follow-up period from follow up).

As suggested by the reviewer, we have conducted a sensitivity analysis and reported our findings as part of the supplementary material (Lines 265-266 and 375-377; and with results tabulated in supplementary file A)

Table 1: Report the statistical test used to produce the P values.

This has now been specified under statistical methods (Lines 262-264).

Lines 306-309: It would be helpful to readers to clarify in the text to which exercise group each number refers.

We thank the Reviewer for pointing out this and have specified the corresponding exercise groups in text (Lines 303-308)

Report the number of IHD events in each exposure group, as well as the overall number

As suggested by the reviewer, this information has now been added (Lines 273-274)

Lines 374-381: This paragraph repeats the results described in lines 364-368.

We thank the reviewer for identifying this mistake. This has now been corrected.

Lines 382-384: This text repeats the results described in lines 370-373.

The correction has been made.

Table 3: CI limits should be reported consistently, with 2 decimal places. I appreciate the authors may wish to show that an interval does not quite include one, but there is no different interpretation/conclusion between an interval that ends at 0.998 or 1.00.

We agree with the reviewer, however, it is a common practice to show that the interval does not contain 1.00.

VERSION 2 – REVIEW

REVIEWER	Yap, Jonathan National Heart Centre Singapore
REVIEW RETURNED	12-Jan-2021

GENERAL COMMENTS	The authors have addressed most of my previous comments except one. Currently PM10 and 2.5 are analysed as binary figures, they mention they could not analyse particle levels as quartiles and continuous variables due to limited statistical power. The study population involves >30,000 subjects. There should be enough power for a sensitivity analysis looking at PM10/2.5 in quartiles and continuous variables.
---

REVIEWER	Wright, Neil Oxford University
REVIEW RETURNED	01-Mar-2021

GENERAL COMMENTS	I appreciate the authors' useful response and updated manuscript. The authors have addressed many comments satisfactorily, but in a few cases greater improvements could be made: The authors' explanation of the choice of cut-off for categorising pollution exposure is reasonable. However, dichotomisation of exposure remains a notable weakness of the analyses. High correlation between variables suggests further investigation could be informative, and residual confounding in the original analyses may be a larger problem. The authors say confounders selected were "consistent with previous studies and known to affect the association between air pollution, physical activity and IHD". The authors should clarify if all adjustments were pre-specified, or could be changed by observed associations in this analysis. If observed associations between covariates and air pollution exposure did inform the models, these should be reported and the methods described.
---

	The authors have added a sensitivity analysis excluding participants with "short" follow-up time. Providing the definition of "short" in the main manuscript would be helpful to readers, plus some interpretation of the result in the discussion. The authors have also satisfactorily address many comments by: Explanation for choice of time scale, and use of calendar time in the models. Adding clarification on the limitation of baseline measurement (Line 464). Adding some assessment of the proportional hazards assumption by inclusion of Schoenfeld tests. Adding detail on the statistical tests reported in Table 1. Helpful clarifications to the text (Lines 306-206). Reporting the number of IHD events in each exposure group (Lines 273-274). Removing repetition of results. Table 3 CI limits: I appreciate the authors' explanation.
--	--

VERSION 2 – AUTHOR RESPONSE

Reviewer: 1

Dr. Jonathan Yap, National Heart Centre Singapore Comments to the Author:

R1.1. The authors have addressed most of my previous comments except one. Currently PM10 and 2.5 are analysed as binary figures, they mention they could not analyse particle levels as quartiles and continuous variables due to limited statistical power. The study population involves >30,000 subjects. There should be enough power for a sensitivity analysis looking at PM10/2.5 in quartiles and continuous variables.

*Response: Even though the study cohort involved a large number of individuals the statistical power was limited by the relatively low spatial variability in particle concentration. A sensitivity analysis has however now been added with particle concentrations in tertiles, to be able to assess a possible interaction dose-response for the risk of incident IHD (page 12 row 369 – page 13 row 386).

Reviewer: 2

Dr. Neil Wright, Oxford University

Comments to the Author:

I appreciate the authors' useful response and updated manuscript. The authors have addressed many comments satisfactorily, but in a few cases greater improvements could be made:

R2.1. The authors' explanation of the choice of cut-off for categorising pollution exposure is reasonable. However, dichotomisation of exposure remains a notable weakness of the analyses. High correlation between variables suggests further investigation could be informative, and residual confounding in the original analyses may be a larger problem.

*Response: We understand and have added sensitivity analyses with particle concentrations in

tertiles to be able to assess a possible interaction dose-response for the risk of incident IHD (page 12 row 369 – page 13 row 386).

R2.2. The authors say confounders selected were "consistent with previous studies and known to affect the association between air pollution, physical activity and IHD". The authors should clarify if all adjustments were pre-specified, or could be changed by observed associations in this analysis. If observed associations between covariates and air pollution exposure did inform the models, these should be reported and the methods described.

*Response: We thank the reviewer for this comment. These covariates were a priori specified, and this is now stated in the methods section (row 258, page 7).

R2.3. The authors have added a sensitivity analysis excluding participants with "short" follow-up time. Providing the definition of "short" in the main manuscript would be helpful to readers, plus some interpretation of the result in the discussion.

*Response: We agree. The term short has now been defined in both methods (page 7, row 269) and results (page 12, row 369), and added to the discussion on the risk of reverse causation (page 16, rows 483-484).

The authors have also satisfactorily address many comments by:

Explanation for choice of time scale, and use of calendar time in the models.

Adding clarification on the limitation of baseline measurement (Line 464).

Adding some assessment of the proportional hazards assumption by inclusion of Schoenfeld tests.

Adding detail on the statistical tests reported in Table 1.

Helpful clarifications to the text (Lines 306-206).

Reporting the number of IHD events in each exposure group (Lines 273-274).

Removing repetition of results.

Table 3 CI limits: I appreciate the authors' explanation.

Reviewer: 1

Competing interests of Reviewer: None declared

Reviewer: 2

Competing interests of Reviewer: None declared